# The Pleiotropy of *PAX5* Gene Products and Function

**DOI:** 10.3390/ijms231710095

**Published:** 2022-09-03

**Authors:** Parinaz Nasri Nasrabadi, Danick Martin, Ehsan Gharib, Gilles A. Robichaud

**Affiliations:** 1Département de Chimie et Biochimie, Université de Moncton, Moncton, NB E1A 3E9, Canada; 2Atlantic Cancer Research Institute, Université de Moncton, Moncton, NB E1C 8X3, Canada

**Keywords:** *PAX5*, BSAP, B-cell, differentiation, maturation, cancer, oncogene, tumor suppressor, leukemia, lymphoma, breast cancer, transcription factor

## Abstract

PAX5, a member of the Paired Box (PAX) transcription factor family, is an essential factor for B-lineage identity during lymphoid differentiation. Mechanistically, PAX5 controls gene expression profiles, which are pivotal to cellular processes such as viability, proliferation, and differentiation. Given its crucial function in B-cell development, *PAX5* aberrant expression also correlates with hallmark cancer processes leading to hematological and other types of cancer lesions. Despite the well-established association of *PAX5* in the development, maintenance, and progression of cancer disease, the use of *PAX5* as a cancer biomarker or therapeutic target has yet to be implemented. This may be partly due to the assortment of *PAX5* expressed products, which layers the complexity of their function and role in various regulatory networks and biological processes. In this review, we provide an overview of the reported data describing *PAX5* products, their regulation, and function in cellular processes, cellular biology, and neoplasm.

## 1. Introduction

The Paired Box (*PAX*) gene family encodes nine transcription factors (PAX1–9), which regulate gene expression programs in tissue development [1]. Although PAX transcription factors share a highly similar paired-box DNA-binding domain, they are classified into four subgroups (I–IV) based on additional functional domains such as the octapeptide and the homeodomain, which are generally located in the protein’s internal and amino-terminal regions respectively [2,3]. Given their structural resemblance, *PAX* members from a particular subgroup account for similar activities and functions. For example, *PAX* genes in subgroups II (*PAX2*, *PAX5*, and *PAX8*) and III (*PAX3* and *PAX7*) are commonly involved in processes including cell survival, motility, and tumor progression. Conversely, members from subgroup I (*PAX1* and *PAX9*) and IV (*PAX4* and *PAX6*) seem less involved in cancer processes [4]. The expression of *PAX* family gene products is also generally tissue specific. For instance, *PAX2* expression has been described in kidney and optic nerve development [5], whereas *PAX5* has mostly been associated with the development of the central nervous system, of B-lymphocytes, and spermatogenesis [6]. Furthermore, the expression patterns of subgroup II members are reported to be altered in various cancer tissues, which suggests a distinctive role for these *PAX* gene products in the regulation of specific malignancies [1]. Amongst these members, *PAX5* has been extensively studied and characterized for its role in cancer pathogenesis.

This review will summarize the upstream and downstream pathways associated with the regulated expression of various *PAX5* gene products in multiple tissues. Specifically, we will describe the most prevalent roles of *PAX5* in vital biological processes (e.g., B-cell maturation) and neoplasia (e.g., leukemia and lymphoma) upon its aberrant expression. We will also review the mechanisms and impact of spatiotemporal regulated expression of *PAX5* in both healthy and malignant cellular settings. Most importantly, we will provide a detailed overview of *PAX5’*s remarkable capacity to generate a range of gene products, which accounts for its pleiotropic function in an array of cell types and tissues (Figure 1).

## 2. The PAX5 Transcription Factor and B-Cell Development

### 2.1. Expression and Tissue Specificity

The human *PAX5* gene locus is located on the 9p13 chromosomal region known to undergo a high degree of alterations leading to its implication in cancer development and progression [22]. Structurally, the *PAX5* gene is characterized by two known distinct promoters, resulting in two alternative transcriptional initiation sites known as *PAX5A* and *PAX5B* [11]. Both transcripts share the same sequence encoded by exon 2 through exon 10. However, they have different sequences in their first exon (1A or 1B), which is dependently linked to their respective promoter regions (*PAX5* 1A versus *PAX5* 1B). Both PAX5A and PAX5B protein variants consist of a 52-kD protein known as the B-cell lineage-Specific Activator Protein (BSAP), which was initially identified as an essential regulator of early B-cell differentiation and commitment [6,23]. Despite their structural similarities, *PAX5A* and *PAX5B* gene products display differential expression signatures and tissue specificity [24,25]. For example, *PAX5A* expression is reported to be mostly restricted to the B-cell lineage, whereas *PAX5B* can also be found in the central nervous system and testis [6,11]. Functionally, although *PAX5A* and *PAX5B* have mutually been described to drive B-cell differentiation, other studies have demonstrated that *PAX5* isoforms 1A and 1B regulate distinct target genes and functions in B-cell models [24,25,26].

### 2.2. Role of PAX5 in B-Cell Lineage Commitment and Maturation

Differentiation of common lymphoid progenitors (CLPs) into immunoglobulin (*Ig*) producing plasma cells is a complex spatiotemporal process where each developmental step of B-lymphocyte maturation requires controlled signaling through specific expression profiles of transcription factor associated with B-cell identity [27]. *PAX5* has been well established as the central coordinator of B-cell commitment and maturation.

Initially, hematopoietic stem cells differentiate in CLPs in the bone marrow through the activity of Fms-Like Tyrosine kinase-3 [28,29]. During early B-cell differentiation, pre-B-cells proliferate under interleukin-7 activation, which is secreted from the stromal environment [30]. During this time, CLPs initiate V(D)J recombination of the immunoglobulin heavy-chain (*IgH*) and light-chain (*Igk* or *Igl*) gene loci to assemble the pre-B-cell receptor (BCR) complex (reviewed in [31]). Subsequent B-cell activation enables the induction of the remaining BCR components, which synergistically function to induce *PAX5* expression and other B-cell phenotypic markers such as the Cluster Differentiation-19 (CD19) and the CD79a (*MB-1* gene) transmembrane proteins involved in B-cell activation through the BCR complex [32,33,34,35]. *PAX5* also functionally cooperates with other transcription factors including, PU.1, B-cell Linker, IKAROS Family Zinc Finger-1 (IKZF1), Early B-cell Factor (EBF), and Transcription Factor-3 (TCF3) to support *PAX5* activity during early B-cell identity and maturation [36,37,38]. Further development of the BCR induces the expression and maturation of surface *IgM* and allows immature B-cells to infiltrate peripheral circulation for eventual antigen-dependent developmental stages of B-cell maturation [39]. Upon encountering antigens by way of the BCR, B-cells migrate to peripheral lymphoid organs for further *Ig* rearrangement to produce high-affinity antibody isotypes. Through this, *PAX5* stimulates the Activation-Induced cytidine Deaminase (*AID*) gene, which is essential for somatic hypermutation and antibody class switching as part of B-cells’ capacity to diversify its *Ig* repertoire [40,41]. Mature and activated B-cells can then differentiate either into short-lived antibody-secreting plasma cells or germinal center memory B-cells for immunological memory, depending on the signaling and requirements of the extracellular environment.

In addition to the induction of pro-B lineage specific genes, *PAX5* also concomitantly suppresses B-lineage inappropriate genes [35,42]. Accordingly, *PAX5* inhibits the expression of Fms-Like Tyrosine kinase-3 (mediator of CLP multipotency), *NOTCH* receptor-1 (mediator of T-cell differentiation) in addition to plasma cell differentiation factors B-cell Induced Maturation Protein-1 (*BLIMP-1*) and X-Box Binding Protein-1 [43,44,45]. However, once all *PAX5*-mediated early development checkpoints are achieved, coordinated attenuation of *PAX5* expression through *BLIMP-1* is implemented to pursue B-cell maturation. Accordingly, many studies have established a bidirectional negative regulatory loop between *PAX5* and *BLIMP-1* where the activation of *BLIMP-1* and concomitant suppression of *PAX5* represent a critical step in plasma cell differentiation [43,45,46,47]. Although recent studies demonstrate that complete *PAX5* repression is not essential for developing plasma cells and antibody secretion, it is required for optimal high-affinity *IgG* production and sustained accumulation of long-lived plasma cells [48].

Altogether, *PAX5* not only mediates B-cell identity, it also concomitantly blocks the differentiation of other (non-B) hematopoietic specification. In fact, studies have shown that *PAX5* is required for the progression of B-cell development beyond the pro-B-cell stage as the suppression of *PAX5* during early B-cell development enables the reversal of B-lineage commitment fate [49,50]. Interestingly, these latter *PAX5*-silenced cells can initiate myeloid differentiation until *PAX5* expression is restored [50]. Furthermore, the potential for B-cell retro-differentiation upon ectopic modulation of *PAX5* expression appears unique to early developmental stages in uncommitted mature B-cells. Accordingly, experiments conducted by Proulx et al. (2010) show that restoration of *PAX5* expression in terminally differentiated cancer B-cells (i.e., multiple myeloma) induces apoptosis of myeloma cells [51]. These findings highlight the impact of *PAX5* and its involvement in various pathways leading to B-lineage commitment, maturation, and function. *PAX5* expression is therefore stringently regulated during B-cell development. However, aberrant expression of *PAX5* also consequently leads to pathological disorders, notably cancer.

## 3. *PAX5* and Cancer

### 3.1. PAX5 Is a Key Driver of B-Cell Malignancies

Given its pivotal regulatory role in B-cell development, *PAX5* also represents a potent oncogene in hematological cancers, particularly lymphoma and lymphocytic leukemia. These findings have been validated by numerous studies showing a direct link between *PAX**5* and the onset of B-cell cancer lesions using various B-cell lines and in murine models [50,52]. In fact, *PAX5* expression is associated with the most common non-Hodgkin B-cell malignancies including: diffused large B-cell lymphoma (DLBCL) [53,54]; chronic lymphocytic leukemia [55]; and B-cell acute lymphoblastic leukemia (B-ALL) [55]. In most cases, deregulated *PAX5* expression impedes the progression of B-cell differentiation leading to hallmark cancer features such as proliferation, apoptosis, and phenotype transitioning processes [56]. Genetic analyses demonstrate that aberrant *PAX5* expression and function in most B-cell malignancies are caused by genetic instability of the *PAX5* locus. For example, somatic mutations of the *PAX5* enhancer region are prevalent in DLBCL, follicular lymphoma, Burkitt’s lymphoma, and mantle cell lymphoma [55,57,58,59]. *PAX5* is also prone to recurrent chromosomal rearrangements, which represent the causal root for specific subsets of aggressive B-cell cancer lesions, notably B-ALL [60,61].

Acute lymphoblastic leukemia is the most common hematological (blood) cancer in children, where the B-cell subtype (B-ALL) is the prevalent form. B-ALL is characterized by recurrent genetic changes that suppress developmental stages beyond B-cell precursors, thus preserving the self-renewal phenotype of non-differentiated cells [7,62,63,64]. It is therefore not surprising that most B-ALL cases are associated to genetic alterations from genes governing the earliest stages of B-lymphoid specification (i.e., *TCF3*, *EBF*, *IKZF1*, and *PAX5*) [38,65,66]. Over 80% of gene lesions found in childhood B-ALL are associated with the *PAX5* and *IKZF1* genes alone [7,67]. A proposed mechanism recently described by Chan et al., (2017) suggests that *PAX5* and *IKZF1* mutations disrupt adequate regulation of downstream metabolic-related genes such as Insulin Receptor, Glucose Transporter-1 (*GLUT1*), Glucose Transporter-6, Glucose-6-Phosphate Dehydrogenase and Hexokinase-2, all of which encode proteins governing glucose uptake and utilization [68]. *PAX5* and *IKZF1* are commonly depicted as metabolic gatekeepers by conferring a chronic state of energy deprivation in hematopoietic stem cells, which limits the energy supply necessary for oncogenic transformation [68]. Aberrant glucose uptake and ATP overproduction are the main components fueling tumor development and oncogenic growth [69,70]. In addition, the overexpression of *GLUT1*, a major glucose transporter, contributes to the maintenance of cancer cell metabolism by increasing glycolysis and energy production [71]. Consequently, dominant negative mutants of *PAX5* and *IKZF1* found in the majority of B-ALL lesions alleviate glucose and energy restrictions, which foster malignant transformation [68]. These findings are also corroborated by Kurimoto et al. (2017) who observed significant increases in *GLUT1* expression following *PAX5* silencing in head and neck squamous cell carcinoma [72]. In addition, they suggest that the *GLUT1*-*PAX5* regulatory axis can also be controlled by the Tumor suppressor Protein-53 (*Tp53*), which is upregulated by *PAX5* via a positive feedback loop [73,74,75,76].

More than a third of all B-ALL cases arise solely from *PAX5* genetic alterations, which include deletions, sequence mutations, and translocations to a range of fusion partners [7,77,78]. Concurrently, studies have demonstrated that *PAX5* haploinsufficiency plays an essential role in lymphoblastic leukemia [8,62,79]. Accordingly, insufficient *PAX5* expression due to somatic alterations in high-risk BCR-ABL1-positive (most frequent B-ALL subtype in adults) and Philadelphia-like B-ALL account for 50% of lymphoid blast crises [67,80]. In the case of germinal mutations, inheritance of altered *PAX5* gene sequences predisposes children to a four-fold increased risk of being diagnosed with B-ALL when they have an affected sibling [81]. On the other hand, a growing number of chromosomal translocations in B-ALL regarding *PAX5* are also reported to produce genetic fusions with different gene partners including: Forkhead Box P1 (3p13) [7,82]; Janus Kinase 2 (9p24) [83]; Autism Susceptibility Candidate 2 (7q11) [82]; Elastin (7q11) [8]; *ETS* Translocation Variant 6 (12p13) [7,60,82]; Promyelocytic Leukemia Protein (15q24) [61]; Zinc Finger Protein 521 (18q11) [7]; Chromosome 20 Open Reading Frame 112 (20q11) [82,84]; Ribosome Binding Protein 1 Pseudogene (7p12.1) [84]; Solute Carrier Organic Anion Transporter Family Member 1B3 (12p12) [84]; Additional Sex Combs Like 1 Transcriptional Regulator 1 (20q11.1) [84]; Kinesin Family Member 3B (20q11.21) [84]; Homeodomain Interacting Protein Kinase 1 (1p13) [85]; Pore Membrane Protein of 121 kDa (7q11) [85]; Dachslund Family Transcription Factor 1 (13q21) [85]; and, Bromodomain Containing 1 (22q13.33) [85]. Interestingly, these fusion proteins all contain the 5′end of the *PAX5* coding region (corresponding to the DNA-binding domain) fused with a functional domain of the partner gene. Although these putative chimeric transcription factors retain DNA *PAX5*-specific motif binding, they are characterized by attenuated *PAX5* transcriptional activities or loss of function [10,61,64,84,86]. Mechanistically, some studies have reported that *PAX5* fusion proteins (e.g., *PAX5-ETS Translocation Variant 6*, *PAX5-Forkhead Box P1*, and *PAX5-Elastin*) function as dominant negative factors against their wild-type *PAX5* counterpart [7,8,9,10,87]. Meanwhile, others show that *PAX5* fusion proteins regulate an independent profile of target genes in B-cells [64,88,89]. Altogether, these studies highlight the importance of tightly orchestrated expression of *PAX5* during early B-cell development where insufficient *PAX5* activity leads to B-cell differentiation blockade and uncontrolled proliferation of immature B-cells [7,62,63,64].

In opposition to early-stage B-cell cancer lesions, which are normally characterized with *PAX5* translocations conferring attenuated forms of *PAX5* function, *PAX5* genetic rearrangement in late B-cell developmental stages is mainly associated with increases in *PAX5* expression and activities. For example, the *PAX5* t(9;14) translocation is a genetic rearrangement involving the complete coding region of *PAX5* (chromosome 9), which is relocated under the control of the potent Emµ promoter of the *IgH* gene (chromosome 14) [11,90]. The *PAX5* t(9;14) recombination is by far the most studied and prevalent *PAX5* genetic alteration in non-Hodgkin’s lymphoma subtypes such as lymphoplasmacytic lymphoma (LPL) and DLBCL [11,54,90]. Given that the repression of *PAX5* expression is generally required for adequate B-cell terminal differentiation, its constitutive overexpression caused by the *PAX5*-*IgH* fusion blocks mature B-cell activation, accumulates inactivated mature B-cell populations, and lowers differentiated plasma cell numbers [11,54,90]. Similarly, studies have substantiated these findings by experimentally inserting a *PAX5* minigene into the *IgH* locus (IgHP5ki), which provokes a t(9;14)(p13;q32) rearrangement of germline mutant knock-in mice [91]. Although the development of the IgHP5ki model was reminiscent of human t(9;14) B-cell lymphoma pathology, it harbors a germline rather than a somatic translocation mutation. Hence, forced expression of *PAX5* in the hematopoietic lineage affected overall hematopoiesis in IgHP5ki mice, notably in T-cell development, which led to the development of T-lymphoblastic lymphomas [91]. Altogether, these studies not only demonstrate that t(9;14)-mediated overexpression *PAX5* is an important cause of B-cell lymphoma; but also suggest a particularly harmful role of t(9;14) in the T-lymphoid system [91].

### 3.2. PAX5 in Non-Hematopoietic Cancers

For nearly two decades, the *PAX5* gene has also been revealed to be an imperceptible regulator of cellular processes in various non-hematological tissues and cancers. To obtain a global perspective of relative *PAX5* expression profiles in non-hematological tissues, levels of *PAX5* transcripts and proteins were recovered from data mining, processed, and plotted across a collection of tissue samples (Figure 2).

In contrast to B-cells, the functional role and outcome of aberrant *PAX5* expression in non-hematological cancers are as diverse as the types of tissue that express *PAX5*. For example, Baumann et al. (2004) observed that elevated *PAX5* expression levels correlate with a subset of malignant (N-type) neuroblastoma in comparison to their benign counterpart (S-type) [92]. This study also demonstrates that recombinant expression of *PAX5* in benign neuroblastoma cell models resulted in more invasive cancer phenotypes [92]. Similarly, elevated *PAX5* expression has been associated with pediatric brain tumors (medulloblastomas), where *PAX5* expression positively correlates with cell proliferation and inversely with neuronal differentiation in desmoplastic medulloblastoma [93]. Another study by Kanteti et al. (2009) demonstrated that elevated levels of *PAX5* expression in small-cell lung cancer induced the expression and activation of the *c-MET* proto-oncogene, also known as the Hepatocyte Growth Factor Receptor and a potent regulator of cancer cell motility and angiogenesis [94]. More recently, Dong et al. (2018) have shown that *PAX5* potentiates cisplatin-based systematic chemotherapy resistance of muscle-invasive bladder cancers [95]. Mechanistically, this latter study demonstrates that the *PAX5* transcription factor transactivates *Prostaglandin-Endoperoxide Synthase-2* gene transcription, which promotes bladder cancer pathogenic features [95]. Other reports have also shown a role for *PAX5* in oncogenic or pro-aggressive features in astrocytoma [96,97], lung cancer [94], cervical carcinoma [98], bladder cancer [95], oral carcinoma [99], neuroblastoma [92], and medulloblastoma [93]. In opposition, numerous studies functionally characterize *PAX5* as a tumor suppressor in hepatocellular carcinoma [73,100], breast carcinoma [75,101,102,103], esophageal squamous cell carcinoma [72], retinoblastoma [104], gastric cancer [74,105], Merkel cell carcinoma [106], ovarian cancer [107], and head and neck squamous cell carcinoma [76]. Accordingly, a study by Kurimoto et al. (2017) shows that *PAX5* overexpression inhibits cell proliferation and chemoresistance of esophageal squamous cells [72], whereas reporting from Liu et al. (2011) demonstrates that *PAX5* blocks cell viability and colony formation of hepatocytes [73].

Despite *PAX5* expression prevalence amongst different non-hematopoietic tissues, the upstream mechanisms responsible for its aberrant expression and often opposing impacts between tissue types are not entirely understood. However, many large-scale and functional genomic analyses commonly demonstrate that the tumor suppressive features of *PAX5* are inhibited by promoter hypermethylation events in non-hematological cancers [73,76,100,105,107,108]. Accordingly, the methylation profile of the *PAX5* promoter region has been proposed as a potential tool with clinical applicability for prognostics and diagnostics in the classification of non-hematological cancer subtypes [74,105,106,107]. Another possible explanation has been provided by reports describing *PAX5* regulation through a direct relationship with *Tp53* [73,74,75,76,96]. Yet, the regulation and outcome of *PAX5*/*Tp53* interplay in non-hematological cancer are still conflicted and appear to be tissue type specific. For example, a study by Stuart et al. (1995) demonstrates that *PAX5*-mediated tumorigenesis of primary human diffuse astrocytoma is supported through the direct binding and inhibition of *Tp53* promoter transactivation [96]. In contrast, when adopting the role of a tumor suppressor, *PAX5* has been shown to transactivate *Tp53* expression and activity in gastric cancer [74], breast cancer [75,103], and hepatocellular carcinoma [73]. A study by Guerrero-Preston et al., (2014) further substantiates the antitumor activity of the *PAX5*/*Tp53* axis by demonstrating that *PAX5* promoter hypermethylation in head and neck squamous cell carcinoma results in inadequate *Tp53* activation [76]. This study also shows that *PAX5* promoter methylation status is directly linked to *Tp53* mutational profiles. These findings are significant given the high frequency of *Tp53* mutation in cancer tissues [109] and its association to epigenetic control of *PAX5* expression and function in non-hematological cancer cell fate. 

### 3.3. PAX5 and Breast Cancer

To date, the functional role of *PAX5* expression and function in non-hematopoietic cancers has largely been elucidated in breast cancer models. Studies demonstrate that *PAX5* overexpression in breast cancer cells leads to a decrease in proliferation, colony formation, and migration through the induction of pro-epithelial features [75,101]. Specifically, *PAX5* is shown to regulate hallmark phenotypic transitioning programs known as the epithelial to mesenchymal transition (EMT) during breast cancer cell metastasis and disease progression [75,101]. This phenotypic plasticity is widely accepted as a multistep biological program enabling breast cancer cells to successfully navigate through the various microenvironments confronted during the metastatic journey [110,111,112]. Generally, EMT initiates metastasis by fostering phenotypic change where mammary epithelial cells gain invasive properties (e.g., anoïkis resistance and migration) to infiltrate the circulatory system and distant organs. Thereafter, reversal of EMT or mesenchymal to epithelial transition (MET) would enable circulating metastatic cancer cells to reboot an epithelial program (e.g., adherence and intercellular tight junctions) to colonize metastatic tumor niches [111,112,113]. A study by Vidal et al. (2010) has demonstrated that *PAX5* promotes pro-epithelial dominant features in breast cancer cells and thereby attenuates malignancy and disease progression in murine models [75]. In parallel, Benzina et al. (2017) have shown that *PAX5* not only promotes breast cancer epithelial identity but also induces MET of aggressive breast cancer cells through direct transactivation of E-cadherin, a pivotal regulator of epithelialization [114]. In fact, E-cadherin is not only a predominant surface phenotypic marker of epithelial cells, but also a key regulator of anti-invasive properties and MET. Surface expression of E-cadherin results in the direct suppression of pro-mesenchymal gene expression (e.g., *SNAIL* [115], *TWIST* [116], *SLUG* [117], and *Zinc Finger E-Box Binding Homeobox 1* [118]) deployed during EMT and disease progression [119,120,121].

Altogether, given its pro-epithelialization role in breast tumors, *PAX5* suppresses invasive properties leading to better prognostic value with a lower risk of disease progression and relapse as long as cancer cells remain in their primary tumor sites (in situ) [102]. However, *PAX5* expression during the shifts of phenotypic programs (EMT and MET) necessary for successful metastasis may also produce dichotomous outcomes for breast cancer disease. For instance, *PAX5* may trigger MET in circulating cancer cells thus implementing epithelial features essential for the establishment of distant metastatic colonization [102]. Accordingly, a study by Ellsworth et al. (2009) demonstrates that relative *PAX5* expression levels are 100-fold greater in metastasized breast cancer cells located in patient lymph nodes in comparison to levels found in their primary tumor counterparts [122]. These observations are largely reminiscent of the paradoxical role of *PAX5* during the early and late stages of B-cell development. As discussed previously, disturbance of *PAX5* spatial and temporal regulated expression ultimately leads to aberrant cellular processes and cancer. In the following sections, we will discuss the potential mechanisms of deregulated *PAX5* expression in addition to the diversity of *PAX5* products, which ultimately determine cellular function and cancer outcome.

## 4. *PAX5* Expression and Regulation

As depicted in Figure 3, *PAX5* is widely associated with various cellular processes and pathologies (Figure 3). Given the essential role of *PAX5*-mediated transactivation of vital genes for cell biology, its deregulation will have consequences on basic cellular processes such as differentiation, viability, and proliferation (reviewed in [4,40,56]). Investigation of deregulated mechanisms leading to aberrant *PAX5* expression and activity is therefore relevant and warranted to provide more insight into the overall comprehension of *PAX5* mechanisms of action. Although the literature provides abundant research characterizing *PAX5*-mediated pathways and interactions, the upstream mechanisms regulating *PAX5* expression are much less defined.

### 4.1. PAX5 Epigenetic Regulation

Many genomic studies have described the *PAX5* locus as a genetic hot-spot susceptible to structural variation [57,58,59,123,124]. For example, *PAX5* expression and function are altered by various genetic alterations, including somatic mutation, translocation, and duplication/polyploidy [10,11,62,78,83]. In addition to genetic mutation, which changes both the transcriptional levels and protein sequences, genes are also submitted to epigenetic deregulation, which impacts overall expression levels [125]. These epigenetic processes include methylation of 5′-cytosine-phosphate-guanine-3′ (CpG) islands, chromatin remodeling via histone modifications, and various RNA-mediated mechanisms, which involve regulatory non-coding RNAs [125,126]. A brief description of each regulatory mechanism and its impact on *PAX5*-mediated function is discussed below.

First, methylation of CpG islands to form 5′-methylcytosine (5mC) is a well-described mechanism to repress transcriptional expression of unwanted genes during fundamental cellular processes such as development and differentiation [127,128,129]. DNA methylation is catalyzed by a group of DNA methyltransferase (DNMT) enzyme members (e.g., DNMT1, DNMT3a, and DNMT3b) [130,131]. DNA methylation can also be reversed by demethylation, which is mediated by Ten-Eleven Translocation (TET) family dioxygenase enzymes, which include TET1, TET2, and TET3 [132]. In fact, B-lineage development is coordinated by the well-timed deployment of B-cell fate transcription factors, which are regulated by epigenetic events and post-transcriptional modifications [133,134]. For example, DNMT1, DNMT3a, and DNMT3b are required for the maturation of hematopoietic stem cells into CLPs, whereas DNMT1 is particularly essential for pre-B-cell differentiation to immature B-cell [128,135]. Subsequent studies have since demonstrated that TET function is required for developing B-cells to transit from the pro-B to pre-B developmental stage [136]. Mechanistically, the B-cell-specific *MB-1* (CD79a) promoter is known to be hypermethylated during hematopoietic stem cells transition to CLPs and then progressively demethylated during the expression and assembling of the BCR components. These events upregulate *PAX5* expression and concomitant target genes to achieve B-lineage identity [63,137,138,139]. On the other hand, attenuation of *PAX5* expression during terminal B-cell differentiation is reported to partly mediated by methylation of *PAX5* [140]. In support of these events, a study by Danbara et al., (2002) demonstrates that genomic demethylation using 5-aza-2′-deoxycytidine in myeloma cell lines results in the reconstitution of *PAX5* expression and its transcriptional target genes (*CD19* and *MB**-1*) [140]. Although the regulation of the complex networks of epigenetic modifications governing B-cell differentiation is only partially understood, one aberrant mechanism leading to deregulated *PAX5* methylation has been described for the inadequate function of *AID* [41,141]. The *PAX5/AID* pathway is essential for somatic hypermutation and antibody class switching during *Ig* production [142]. However, constitutive expression of *AID* has been associated with lymphomagenesis through its capacity to alter the sequence of non-*Ig* genes (i.e., *PAX5*) or through *AID*-mediated deamination of the *PAX5* gene [141,143]. As a result, changes in *PAX5* gene sequences redefine motif-specific regions marked for epigenetic modifications and subsequent expression control [41,141].

Given the importance of adequate methylation processes regulating *PAX5*-induced B-cell development, deregulated methylation results in the destabilization of B-cell homeostasis and cancer phenotypes [139,144]. This phenomenon has been further substantiated by the demonstration that *PAX5* methylation status directly correlates with overall survival rates of cancer patients [72,74,105]. Furthermore, studies profiling methylation signatures in pediatric ALL patients have correlated *PAX5* hypermethylation to the pathogenesis of B-ALL and T-ALL subtypes [145,146]. These findings have also prompted Nordlund et al. (2015) to propose that *PAX5* methylation status combined with the mapping of *PAX5* gene recombinations with other partner genes represent an effective diagnostic tool to classify heterogeneous and cytogenetically undefined ALL subtypes [147].

*PAX5* aberrant methylation is not a tissue-specific phenomenon. In fact, *PAX5* hypermethylation has been described in many non-hematological cancers, particularly where *PAX5* is characterized as a tumor suppressor (e.g., hepatocellular carcinoma [100], ovarian carcinoma [107]; head and neck cancer [76], gastric cancer [105], lung and breast cancer malignancies [108,148]). Mechanistically, many of these latter studies demonstrate that silencing of *PAX5* expression by hypermethylation leads to the inadequate transactivation of *Tp53* expression, thus ensuing uncontrolled proliferation or decreased chemosensitivity to anticancer treatment regimens [72,73,149].

Gene expression profiles are also epigenetically regulated by multiple histone-modifying enzymes, which change chromatin structure to alter promoter region accessibility and recruit other modifications [150]. Histones, which assemble the nucleosomes, are prone to modifications, which include acetylation, methylation, ubiquitination, phosphorylation, and sumoylation [150]. The most common modifications consist of arginine methylation and/or lysine acetylation, where acetylation generally promotes gene expression whereas methylation elicits the opposite effects. Many histone modifying enzymes have been characterized including histone acetyltransferases (HATs), histone deacetylases (HDACs), histone demethylases, and various methyltransferases (e.g., Euchromatic Histone-Lysine N-Methyltransferase-2/EHMT2 and Lysine Methyltransferase-2A [128,144]). Like CpG island methylation, chromatin modifications represent an intrinsic part of B-cell activation and differentiation. For example, during early B-cell development, *PAX5* secures B-cell commitment through activating B-cell specific genes. In addition, *PAX5* concomitantly inhibits B-lineage inappropriate genes through the recruitment of HDACs to modify and silence promoter activation of these genes [139]. Studies show that the *PAX5* locus is also continuously regulated by histone modifications throughout B-cell maturation. Specifically, the *PAX5* promoter in pro-B-cells are modified by HDACs, whereas EHMT2 regulates mature B-cells located in germinal centers [128]. Another example is the EBF transcription factor, which is shown to be implicated in *PAX5* and *CD19* transactivation through the silencing of Lysine Methyltransferase-2A during early B-cell development [134,151,152]. Another example is the previously mentioned *PAX5*/*BLIMP-1* axis during terminal B-cell differentiation into plasma cells. It is reported that *BLIMP*-*1* suppresses *PAX5* expression through the recruitment of histone demethylases and EHMT2 activities on the *PAX5* promoter [153,154]. Furthermore, transcription factor Forkhead Box Protein-O1, which is essential for B-cell development beyond the pro-B-cell stage [155], is only activated upon histone methylation of *TCF3*, which only then can elicit histone modifications and silence *PAX5* to enable the progression of B-cell development [156]. Another study conducted by Danbara et al., (2002) has specifically demonstrated that the upstream *PAX5* promoter (exon 1A) is predominantly inactivated by DNA methylation, whereas the downstream promoter (exon 1B) is repressed by histone deacetylation during the final stages of B-cell terminal differentiation [140]. Comprehensively, deregulation of histone modifying events on *PAX5* or its upstream regulators lead to aberrant *PAX5* transcript levels and the development of diseases [144]. Accordingly, a recent study from Jin et al. (2021) has not only shown that *PAX5* is hypermethylated in retinoblastoma tumors but also, the treatment of patients with cyclophosphamide (a common antineoplastic agent to treat retinoblastoma) increases *PAX5* expression via gene demethylation and concomitant DNMT inhibition, which result in tumor regression [104,157].

To add complexity and appreciation for epigenetic mechanisms, different ATP-dependent chromatin remodeling complexes (CRC) capable of moving, ejecting, or restructuring nucleosomes (events often associated with DNA repair) have also been associated with *PAX5* regulation and function [138,158]. For example, SWItch/Sucrose Non-Fermentable and the Nucleosome Remodeling Deacetylase CRCs are known to mediate *PAX5*-dependant induction or repression respectively of *MB-1* (CD79a) gene expression during BCR assembly [138]. Therefore, the opposing functions of CRCs provide another layer of *PAX5* function during B-cell development [138]. Another example is the histone modifying enzymes HATs, which can acetylate other cellular proteins (e.g., transcription factors) besides histones. A study by He et al., (2011) has found that histone acetyltransferase E1A binding protein p300 interacts with the C-terminal region of *PAX5* to acetylate multiple lysine residues of the paired box DNA binding domain [19]. They also demonstrate that acetylation of the *PAX5* transcription factor dramatically enhances the transactivation potential of its target genes [19]. This interaction was also investigated in B-cell lymphoma, where the Metastasis-Associated Protein-1 represents a substrate for acetylation upon its interaction with the HAT p300 [159]. This study found that Metastasis-Associated Protein-1 acetylation leads to the direct transactivation and overexpression of *PAX5*, a widespread phenomenon in human DLBCL [159].

The final contributing mechanism in epigenetic control is mediated by non-coding RNAs, which include small interfering RNAs, microRNAs (miRNAs), piwi-interacting RNAs, long non-coding RNAs, and circular RNAs (circRNAs) [160,161,162,163]. In comparison to DNA and histone modifications, only a paucity of studies has directly elucidated ncRNA-mediated mechanisms governing *PAX5* expression and function. A recent study from Harquail et al., (2019) has used a bioinformatic approach to establish a causal link between differentially expressed miRNAs in cancer cells in relation to their putative targeting of *PAX5*-dependent cancer processes and identified miRs-484 and 210 as directly regulators for *PAX5* expression and function [164]. Interestingly, miR-210 has been extensively studied as a potent oncogenic miRNA, which targets critical tumor suppressors such as *E2F3* and *Tp53* [165,166]. It is also well established that miR-210 is upregulated during hypoxia to induce EMT and tumor progression [167,168,169]. Given the prevalent role of *PAX5* in epithelialization and EMT-MET processes in breast cancer cells [75,101], it has been suggested that miR-210 likely targets *PAX5* during tumor neoplasm and hypoxia to produce a robust, comprehensive shift from epithelial to mesenchymal phenotypic features to evade hypoxic insult [164]. *PAX5* has also been reported to be part of a regulatory feedback loop with miR-155 in cancer cells [170]. MiR-155 is known to play a vital role in the differentiation of memory B-cells where it targets *PU.1* and *AID* necessary for B-cell commitment into plasma cell [171,172]. Despite the rapidly growing field of non-coding RNA function in biological processes, the elucidation of non-coding RNA-dependent control of *PAX5* expression and function in B-cell development and disease is still under investigation. As our knowledge expands on the deregulation of miRNA profiles and its impact on biological processes, we notice that changes to the mRNA sequences targeted by miRNAs will also have significant consequences, including miRNA motif accessibility and disruption of translational control. Accordingly, the next section will discuss *PAX5* post-transcriptional modifications and editing, which alter miRNA-specific targeting and impede the potential binding capacity of any motif-specific interacting partners of *PAX5* products.

### 4.2. PAX5 Post-Transcriptional Regulation

Similar to most human gene transcripts, *PAX5* mRNAs undergo alternative splicing processes, which translate into altered translational reading frames and often multiple protein isoforms [15,16,17,173]. To date, alternative splicing events of *PAX5* transcripts in humans and other species result in translated products with deleted regions corresponding to single or multiple coding exons [15,17,173]. Specifically, studies have shown that alternative splicing of the 5′ or 3′ end of *PAX5* mRNA leads to structural and functional alterations of the *PAX5* transcription factor in the DNA binding (exons 2-3) and transactivation domains (exons 8-9) respectively [17,174,175]. A study performed by Robichaud et al., (2004) has characterized alternatively spliced *PAX5* transcripts in CD19^+^ peripheral blood lymphocytes from healthy adult donors and found that B-cells simultaneously co-express multiple isoforms, including full-length mRNA (exons 1-10), in addition to transcripts lacking either exon 7 (∆7); exon 8 (∆8); exon 9 (∆9); exons 7-8 (∆7/8); or exons 7-8-9 (∆7/8/9) [17]. Interestingly, this study also demonstrates that each *PAX5* protein variant elicits a unique transactivation potential upon downstream target genes [17]. Other studies have since reported additional C-terminal isoforms lacking exons 6-7-8-9 (∆6/7/8/9); exons 6-7-8 (∆6/7/8); exons 8-9 (∆8/9); and finally, a transcript containing a partial intronic sequence (intron 6) in healthy B-cells and lymphoma [175]. These findings underscore the complexity of potential dominant-negative effects and the outcome of downstream target gene expression due to a network of multiple *PAX5* transcription factor variants. Despite various reports characterizing the expression of alternatively spliced *PAX5* variants, the specific role of each isoform and their capacity to compete for putative *PAX5* targets are still undefined. However, one study conducted by Sadakane et al., (2007) has correlated a specific expression profile comprising of the wild-type and the ∆8 *PAX5* variants in over 90% of childhood acute lymphoblastic leukemia samples tested [176]. These findings suggest a possible role for individual *PAX5* alternatively spliced isoforms in the regulation (or deregulation) of *PAX5* function.

More recently, *PAX5* transcripts have also been characterized to undergo 3′end shortening [18]. This type of transcriptional modification has significant repercussions on translational fate given that mRNA untranslated regions (UTRs), notably at the 3′end, harbor multiple binding sites for RNA binding proteins and other translational regulatory elements (e.g., miRNAs), which control transcript stability and translation efficiency [177,178]. A study by Beauregard et al., (2021) has recently reported that although 3′-editing of *PAX5* transcripts is prevalent in healthy peripheral B-cells, shortening of the 3′UTR is directly linked to increased translation of *PAX5* and correlates with leukemic disease progression [18]. Mechanistically, the study reveals that *PAX5* 3′UTR shortening is mainly due to sequence excision (up to 86%) by alternative splicing events. *PAX5* mRNA shortening was also investigated in non-hematological cancers. Interestingly, conversely to 3′UTR splicing in B-cells, *PAX5* 3′UTR shortening in breast cancer cells is primarily manifested by alternative polyadenylation (APA) [18]. APA is another type of post-transcriptional modification where gene transcription is prompted to use alternative polyadenylation motifs (transcription termination signals), which alter the overall length of the mRNA sequences at their 3′ end. In fact, APA motifs are prevalent in more than half of all human transcripts, notably in oncogenes, to evade translational control at their 3′UTR, resulting in increased mRNA stability and translation [179,180,181]. To further elucidate the impact of *PAX5* 3′UTR shortening on miRNA targeting and regulation, a bioinformatic approach was used to identify predicted miRNAs targeting the excised 3′UTR in truncated *PAX5* transcripts from cancer cells [18]. The study then experimentally validated that miR-181a, miR-217, and miR-1275 represent the most impactful tumor suppressors lost during *PAX5* 3′UTR shortening in cancer cell models [18]. Nevertheless, more studies are required to fully understand the impact of regulatory elements (e.g., miRNAs) and the accessibility of their corresponding binding sites deleted from truncated *PAX5* transcripts in oncogenic processes and disease.

The previous sections describe multiple regulatory mechanisms, which lead to very diverse *PAX5* transcripts, proteins, and functions. More recently, we and others have also characterized a new class of transcriptional *PAX5* products, circular *PAX5* RNAs (*circPAX5*) [12,13,14]. Circular RNAs (circRNAs) represent a relatively new category of non-coding RNAs characterized by a covalent phosphodiester bond between the 5′ and 3′ extremities of the transcript [182,183]. After being discovered in viruses in 1976, circular RNA was first observed in humans in 1991, when it was initially thought to be the product of improper post-translational editing [184]. Since then, circRNAs have been shown to be abundantly expressed and play essential roles in cell biology and disease [163,182]. Accordingly, circular RNAs can encode proteins through cap-independent translation pathways, regulate gene transcription, regulate gene translation, interact with proteins, and even mop up (sponge) small RNAs such as miRNAs [185,186,187]. Due to their essential role in cellular processes, circular RNA aberrant expression and function are consequently associated with diseases, including cancer [184,188,189]. In fact, a study by Gaffo et al. (2019) describes *circPAX5* as one of the most differentially overexpressed products in pediatric B-ALL patients [12]. They also demonstrate that *circPAX5* directly binds to miR-124-5p in B-cell precursors to promote B-ALL progression through the interference of the B-cell maturation process [12]. More recently, we have mapped multiple *circPAX5* isoforms in B-cells including: *circPAX5_2-3* (containing exons 2 and 3); *circPAX5_2-4* (exons 2, 3 and 4); *circPAX5_2-5* (exons 2, 3, 4 and 5); *circPAX5_2-6* (exons 2, 3, 4, 5 and 6); *circPAX5_2-7* (exons 2, 3, 4, 5, 6, and 7); *circPAX5_2-8* (exons 2, 3, 4, 5, 6, 7, and 8); *circPAX5_8* (exon 8); *circPAX5_7-8* (exons 7 and 8); *circPAX5_5-8* (exons 5, 6, 7, 8); and finally, *circPAX5_2-6+intron 5* (exons 2, 3, 4, 5, partial intron 5, and exon 6) [13,14]. Furthermore, using TaqMan probes designed to target each unique *circPAX5* junction region created by both extremities, we demonstrate that *circPAX5_2-5* and *circPAX5_2-6* are overexpressed in chronic lymphocytic leukemia patients in comparison to peripheral B-cells from healthy individuals. Mechanistically, we demonstrate that *circPAX5* products interact with important microRNAs such as miR-146a and the miR-17-92 cluster. Previous reports demonstrate that miR-146a is a critical regulator of *BLIMP-1* during B-cell differentiation [190], whereas microRNAs from the miR-17-92 cluster mediate the developmental transition of pro-B to pre-B-cells [191,192]. In addition, the miR-17-92 cluster is also associated with many oncogenic processes and phenotypes of hematopoietic cancers, notably in Burkitt lymphoma [193]. Altogether, these findings not only identify a new class of *PAX5* products (i.e., *circPAX5*) but also provide new potential signaling avenues for *PAX5*-mediated function in B-cell development and disease.

### 4.3. Post-Translational Regulation of PAX5

Post-translational modifications and regulation of PAX5 function have not been extensively characterized. However, a few studies have reported specific PAX5-interacting regulators, which modify the PAX5 transcription factor to regulate its transactivation potential. As described earlier, PAX5 can be acetylated by HATs on multiple lysine residues, which enhances its transcriptional activation of downstream target genes [19]. Another example is how the PAX5 transcription factor can be regulated through phosphorylation events. Accordingly, studies show that PAX5 phosphorylation is responsible for the *BLIMP-1*/*PAX5* regulatory axis during the critical stages of plasma cell differentiation [20,21]. Specifically, upon BCR engagement of pro-B cells, PAX5 is phosphorylated on serine and tyrosine residues by Extracellular Regulated Kinases-1/2 and Spleen Associated Tyrosine Kinase respectively, which revoke PAX5′s ability to repress *BLIMP-1* expression, thus enabling the progression of plasma cell development. On the other hand, a study conducted by Kovac et al., (2000) has demonstrated that Importin alpha-1 interacts with the nuclear localization signal on PAX5 to confer its nuclear localization and import, leading to greater PAX5 transactivation of downstream target genes [194].

## 5. Discussion

It is well established that *PAX5* products are important regulators of cell biology, notably in B-lineage commitment and maturation. It is also apparent that *PAX5* is plagued not only by the high-profile genes it regulates but also, by its incredible vulnerability to genetic alterations leading to aberrant expression of *PAX5* products. Given the reliance of crucial developmental program genes on *PAX5* transactivity, perturbation of *PAX5* expression and function at any level ultimately derails basic cellular processes, lending way to oncogenic manifestations. Moreover, given the requirements for coordinated and transitional *PAX5* expression profiles during early (*PAX5* activation) and late (*PAX5* attenuation) phases of B-cell maturation, inadequate *PAX5* activity leads to blockade of B-cell differentiation and uncontrolled proliferation of immature B-cells [7,62,63,64].

This review first describes how *PAX5* liability is evidenced through its high susceptibility to genetic alterations, which either increase or decrease *PAX5* expression and/or activity depending on the type of rearrangement. For example, impairment of *PAX5* expression homeostasis can be allocated through multiple events, including unsuitable promoter regions (e.g., t(9;14)) [11,90], substitution of *PAX5* protein domains (chimeric proteins) [8,10,60,88], and inadequate DNA motif marking for epigenetic control [140,145]. Although genomic instability has been extensively studied and correlated to aberrant *PAX5* expression in hematological tissues, studies have also ruled out *PAX5* genetic mutation as a causal link for the deregulation of *PAX5* expression and function. In fact, *PAX5* translocations are not well described in non-hematological cancer lesions. Regulated *PAX5* expression and function are therefore highly dependent on post-transcriptional and translational mechanisms, including alternative splicing, use of alternative promoters (1A versus 1B), 3′UTR shortening, RNA circularization, and protein phosphorylation/acetylation, all of which play an essential role in *PAX5* regulation (Figure 4).

*PAX5* post-transcriptional and translational processes not only regulate *PAX5* expression, but also contribute to *PAX5* product diversity. The variety of *PAX5*-generated products likely confer multifaceted functions and influence on biological processes in various tissue types. The assortment of *PAX5* products therefore complexifies our mechanistic elucidation of gene function and may account for the seemingly conflicting reports on *PAX5* function in some cancer processes. Studies have also shown that multiple *PAX5* products are expressed simultaneously within a cellular context where specific variants can elicit unique functions [25,26,176]. Therefore, functional genomics should not only consider the individual role of a particular *PAX5* variant but rather *PAX5* products altogether for a potential cellular outcome. For example, the evaluation of *PAX5* function should consider expressed ratios, neutralizing actions, synergistic effects, or dominant negative events from different *PAX5* products on overall *PAX5*-mediated gene expression programs and cellular processes. Additionally, technical scrutiny is advised during the evaluation and profiling of *PAX5* expression as multiple *PAX5*-derived products have recurring homologous sequences with overlapping similarities. For example, the targeted PCR amplification of *PAX5* exons 5-6 would reveal all expressed RNAs including: *PAX5a* (exon 1A); *PAX5b* (exon 1B); *PAX5* full-length mRNA in addition to alternatively spliced isoforms (e.g., ∆7, ∆7/8, ∆7/8/9, etc.); truncated 3′UTR *PAX5* mRNAs; and circular *PAX5* RNA variants (e.g., *circPAX5*_2-3, *circPAX5*_2-5, *circPAX5*_2-6, etc.). 

Clinically, the high specificity and sensitivity of the PAX5 transcription factor have made it a useful marker in identifying and distinguishing lymphomas and leukemias of B-cell origin. Indeed, some studies has proven *PAX5* expression to be a more specific marker than CD79a for the diagnosis of B-ALL [195]. It is well established that *PAX5* deletion is common in childhood and adult B-ALL, supporting its value in diagnosing or monitoring B-ALL [7,196]. *PAX5* rearrangement is also a potent diagnostic marker where *PAX5* t(9;14) is the most prevalent genetic alteration in lymphoplasmacytoid lymphoma and occasionally DLCL [11,90,197]. Other studies have also suggested the use of *PAX5* recombination profiles can be used in a panel along with other transcription factor coding genes to predict disease outcomes and potential relapses following therapy [198]. Additionally, methylation of *PAX5* has been characterized as a tumor-specific event in head and neck squamous cell carcinoma [72]. These findings have also been supported by others, which propose that *PAX5* methylation status combined with mapping of *PAX5* gene recombinations represent an effective diagnostic tool to classify undefined ALL subtypes [147]. On the other hand, *PAX5* could also represent a potent therapeutic target for many hematological cancer lesions. However, these perspectives will only be achieved upon the careful investigation and functional elucidation of different *PAX5* products. 

Overall, the complexity of *PAX5* products and signaling is growing at many levels. The expression profiles of various *PAX5* products and their vast interacting networks ultimately determine the outcome of cell fate events and/or cancer processes. To this point, further elucidation is warranted to provide more insight into the overall comprehension of *PAX5* function in cell biology and disease.

## Figures and Tables

**Figure 1 ijms-23-10095-f001:**
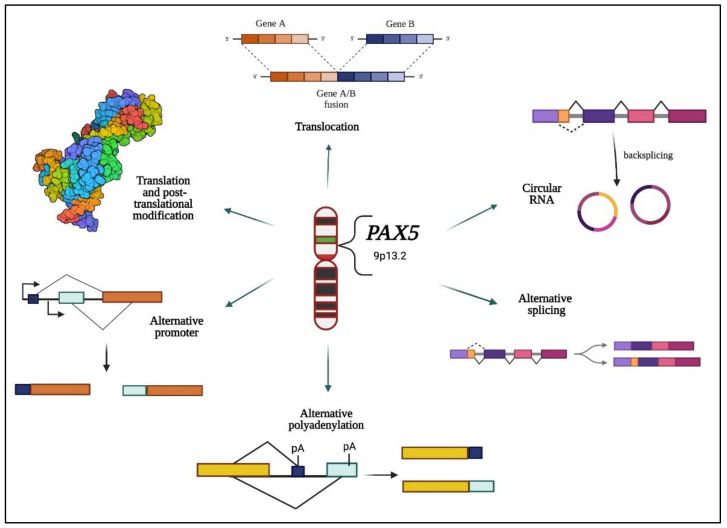
Hallmarks of *PAX5* gene alterations and expressed products. This schematic illustration represents the reported modifications of the *PAX5* gene sequence and expressed products. Multiple studies indicate that the human *PAX5* gene undergoes various alterations that affect its spatiotemporal expression to function as an oncogene or tumor suppressor. Specifically (clockwise from the top), the *PAX5* locus can undergo: genetic translocations resulting in chimeric fusion proteins and alternative promoter regulation [7,8,9,10,11]; transcriptional back-splicing resulting in circular *PAX5* RNAs [12,13,14]; transcriptional alternative splicing of coding exons [15,16,17]; alternative use of poly-adenylation (pA) termination signals [18]; use of alternative promoter regions and transcription initiation sites [11]; and post-translational modifications, which impact its transactivation potential [19,20,21].

**Figure 2 ijms-23-10095-f002:**
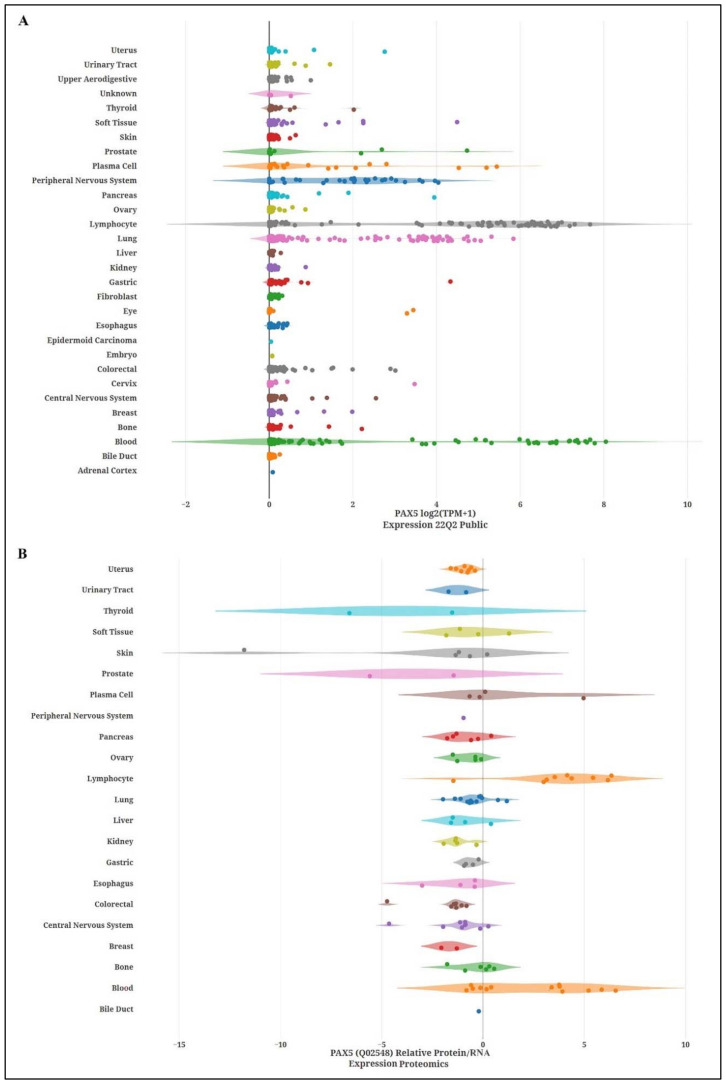
*PAX5* expression profiles in human tissue collections. (**A**) Relative *PAX5* transcription levels from deep sequencing data in over 560 cancer cell lines depict a significant level of *PAX5* RNA expression in human malignancies. The data was plotted using DepMap (https://depmap.org/portal/depmap, accessed on 1 April 2022). Raw reads were aligned with the STAR algorithms (https://github.com/alexdobin/STAR, accessed on 1 April 2022). Relative transcript quantification was performed with the RSEM and subsequently TPM-normalized. Final data were plotted as log2(TPM+1). (**B**) Comparison analysis of *PAX5* protein/RNA levels in different cancer types are plotted and show a significant translation of *PAX5* RNA in blood, bone, and human lymphocytes. In contrast, despite the high RNA expression, *PAX5* protein level remains low in most examined solid cancer tumors, suggesting that these cells use alternative regulatory mechanism affected by *PAX5* RNA products. The data was analyzed using the Proteomics DB web tool (https://www.proteomicsdb.org, accessed on 4 April 2022). Values are shown as median value ± interquartile range in log2-transformed iBAQ intensities.

**Figure 3 ijms-23-10095-f003:**
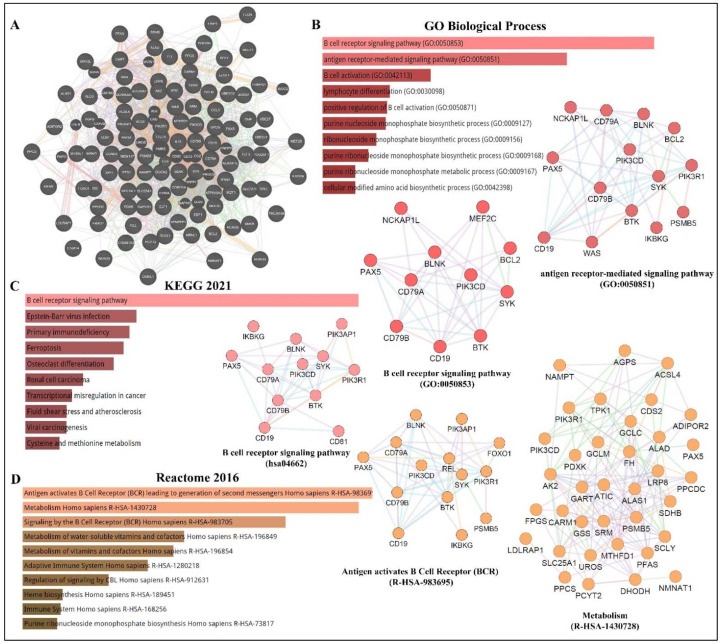
*PAX5* interaction networks and related biological pathways. (**A**) *PAX5* gene interaction networks have been mapped using the Cytoscape plugin GeneMANIA (https://genemania.org, accessed on 8 April 2022). Schematic illustrations of functional annotations and biological terms visualization are represented by: (**B**) *PAX5* gene ontology (GO) in terms of functional orthologs and their relative implication in each predicted biological processes; (**C**) *PAX5* pathway analysis using the Kyoto Encyclopedia of Genes and Genomes (KEGG) database, which provides an integrated evaluation of genomic, chemical, and biochemical functions; and (**D**) relative functional association to biological reactomes based on *PAX5*-related network genes. Annotations were done using the *Enrichr* algorithms (https://maayanlab.cloud/Enrichr, accessed on 10 April 2022). Significance was considered if *p* < 0.05.

**Figure 4 ijms-23-10095-f004:**
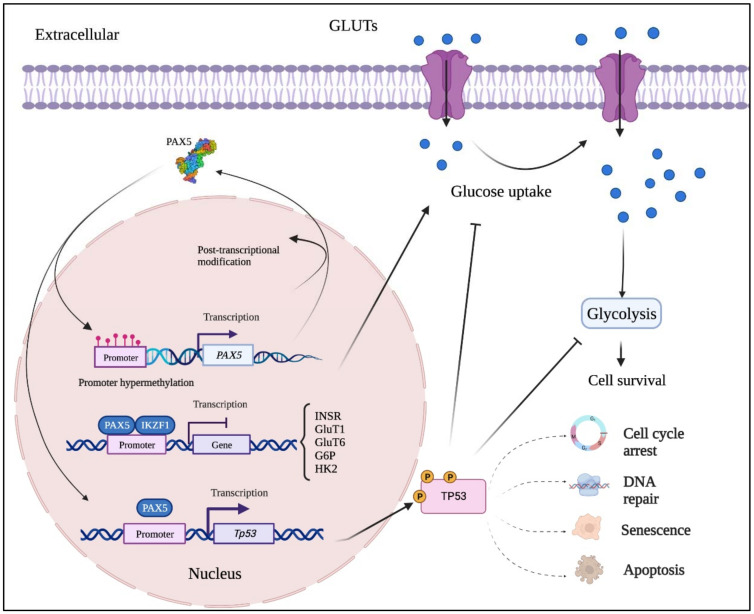
Mutation-independent mechanisms leading to aberrant *PAX5* signaling and cell processes. Aside from *PAX5* gene sequence alterations, deregulated *PAX5* expression can also result from epigenetic events and post-transcription modifications. First, *PAX5* gene promoter hypermethylation has been described in many cancers, notably when *PAX5* behaves as a tumor suppressor. Post-transcriptional modifications (e.g., coding exon alternative splicing, 3′UTR shortening, and RNA circularization) also contribute to overall *PAX5* expression and function. The net production of functional *PAX5* transcription factors can thereafter collaborate with *IKZF1* to regulate downstream metabolic genes to limit glucose uptake and energy supply required for oncogenic transformation. Adequate *PAX5* function is also required to regulate *Tp53* expression and avoid uncontrolled cancer phenotypes. *Tp53* is also intimately linked to metabolic disfunction leading to cancer processes.

## Data Availability

The illustrated transcriptional data from Figure 2 was obtained from DepMap (https://depmap.org/portal/depmap, accessed on 1 April 2022). Raw reads were aligned with the STAR algorithms (https://github.com/alexdobin/STAR, accessed on 1 April 2022). Data and comparison analysis of *PAX5* protein levels were obtained from the Proteomics DB web tool (https://www.proteomicsdb.org, accessed on 4 April 2022). The illustrated transcriptional data from Figure 3 was obtained from the Cytoscape plugin GeneMANIA (https://genemania.org, accessed on 8 April 2022). Schematic illustrations of functional annotations and biological terms visualization were done using the Enrichr algorithms (https://maayanlab.cloud/Enrichr, accessed on 10 April 2022).

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
