# Peer review of "The Pleiotropy of PAX5 Gene Products and Function"

_ijms, 2022, doi:10.3390/ijms231710095_

Round 1
Reviewer 1 Report
The manuscript entitled "The pleiotropy of Pax5 gene products and function" is an extensive review of the knowledge that has been acquired on the role during development and in cancer of Pax5 and its mechanisms of action. The structure of the article is logical and follows a plan that goes from a general introduction placing this gene in the family from which it is derived, then addresses in the following order i) the expression and physiological functions of Pax5, ii) its role in cancer and iii) the molecular regulatory mechanisms modulating its activity. Finally, a discussion closes the review and provides insights into areas of gene function regulation that are still under-researched, such as those of circular RNAs.
The review is captivating and reads very well. I have a few minor comments:
A) The font of letters in the Figures, especially Figure 2 and 3, are too small; the figures lose readability.
B) I wonder if the title "the PAX5 transcription factors and Function" is completely appropriate, more contrast with the following paragraph is desirable.
C) The nomenclature of genes and proteins should follow international recommendations.
D) The number of abbreviations is large; it is sometimes difficult for researchers outside the field of hemapoiesis to follow. When abbreviations appear only twice, they are not necessary.
E) There are some mistakes here and there. Figures that become figures. Not many.
Author Response
"Please see the attachment."

Reviewer 2 Report
Authors provide an excellent review of Pax5 expression, gene products, and their relationship to cancer both as an oncogene and as a tumor suppressor. More specifically, they provide a detailed description of Pax5 expression and the role of Pax5 in most malignant B-cell tumors.
They also report Pax5's role as a potent oncogene in hematologic cancers, specifically lymphoma and lymphocytic leukemia, partially through the resulting dysregulation of metabolism-related genes that promote tumor development and oncogenic growth.
In addition, the authors highlight the alterations that cause Pax5 to function as an oncogene or tumor suppressor, promoting tumors such as desmoplastic medulloblastoma, astrocytoma, lung cancer, cervical carcinoma, bladder cancer, oral carcinoma, or neuroblastoma. Although it also functions as a tumor suppressor through methylation processes in hepatocellular carcinoma, breast carcinoma, esophageal squamous cell carcinoma, retinoblastoma, gastric cancer, Merkel cell carcinoma, ovarian cancer, and head and neck squamous cell carcinoma.
Finally, the authors describe the epigenetic regulation of Pax5 by methylation or by chromatin remodeling by histone modification, as well as the products of Pax5 regulation and posttranscriptional modification.
I suggest publishing it without further changes.
Author Response
We thank you for your time and consideration in reviewing our proposed manuscript.
Reviewer 3 Report
To date, the expression of different Pax5 products and their signaling pathways gain great attention due to the effect on cell fate events and/or cancer development. the authors in the review entitled “The pleiotropy of Pax5 gene products and function” highlighted the description of Pax5 products, their regulation, and function in cellular processes, cellular biology, and neoplasm. The manuscript is well written and designed. However, there are a few comments for the authors to work on.
1- The authors reported in “Role of Pax5 in B-cell lineage commitment and maturation” section that “Interestingly, these latter Pax5-silenced cells can initiate myeloid differentiation until Pax5 expression is restored”, is it “can” or “can’t”?
2- The authors should introduce more readable data for Figure 2. “Human Pax5 expression profiles in human tissue collections.”
3- The authors should revise the manuscript for the typos along the manuscript ex: In Pax5 epigenetic regulation section “, which only then can elicit histone modifications an silence Pax5 to enable the progression of B-cell development”, etc.
4- I do suggest the authors should highlight the clinical applications of the Pax5 genes and their future perspectives in a separate section or in the discussion part.
Author Response
"Please see the attachment."
